# Nanoparticle Size and Heat Pipe Angle Impact on the Thermal Effectiveness of a Cylindrical Screen Mesh Heat Pipe

**Prabhu Alphonse** [1], **Karthikeyan Muthukumarasamy** [1] **and Ratchagaraja Dhairiyasamy** [2,*]

[1] Department of Mechanical Engineering, Faculty of Engineering and Technology, Annamalai University, Tamil Nadu 608002, India; prabhualphonse@gmail.com (P.A.); karthi3152@gmail.com (K.M.)

[2] Department of Mechanical Engineering, College of Engineering and Technology, Aksum University, Aksum P.O. Box 1010, Ethiopia

* Correspondence: ratchagaraja@gmail.com

**Abstract:** This study examines the effects of particle size and heat pipe angle on the thermal effectiveness of a cylindrical screen mesh heat pipe using silver nanoparticles (Ag) as the test substance. The experiment investigates three different particle sizes (30 nm, 50 nm, and 80 nm) and four different heat pipe angles (0°, 45°, 60°, and 90°) on the heat transmission characteristics of the heat pipe. The results show that the thermal conductivity of the heat pipe increased with an increase in heat pipe angle for all particle sizes, with the highest thermal conductivity attained at a 90° heat pipe angle. Furthermore, the thermal resistance of the heat pipe decreased as the particle size decreased for all heat pipe angles. The thermal conductivity measurements of the particle sizes—30, 50, and 80 nm—were 250 W/mK, 200 W/mK, and 150 W/mK, respectively. The heat transfer coefficient values for particle sizes 30 nm, 50 nm, and 80 nm were 5500 W/m²K, 4500 W/m²K, and 3500 W/m²K, respectively. The heat transfer coefficient increased with increased heat pipe angle for all particle sizes, with the highest heat transfer coefficient obtained at a 90° heat pipe angle. The addition of Ag nanoparticles at a volume concentration of 1% reduced the thermal resistance of the heat pipe, resulting in improved heat transfer performance. At a heat load of 150 W, the thermal resistance decreased from 0.016 °C/W without nanoparticles to 0.012 °C/W with 30 nm nanoparticles, 0.013 °C/W with 50 nm nanoparticles, and 0.014 °C/W with 80 nm nanoparticles. This study also found that the heat transfer coefficient increased with increased heat pipe angle for all particle sizes, with the highest heat transfer coefficient obtained at a 90° heat pipe angle.

**Keywords:** heat pipe; particle size; heat transfer; thermal conductivity; heat transfer coefficient

## 1. Introduction

Heat transfer is critical in various industries, including aerospace, automotive, electronics, and power generation. Heat pipes are widely used in these industries due to their efficient heat transfer capabilities [1]. Heat pipes transfer heat from a heat source to a heat sink using a working fluid, typically a liquid or gas. The working fluid's thermal conductivity is a key parameter affecting the heat pipe's heat transfer effectiveness [2].

Nanoparticles in heat pipes have been shown to improve their thermal conductivity significantly. Adding nanoparticles to the working fluid can enhance heat transfer by providing a large surface area for heat transfer and increasing the effective thermal conductivity of the fluid. The size of the nanoparticles can also significantly impact the heat transfer effectiveness of the heat pipe [3].

Nanofluids have become an active research area due to their unique thermal and physical properties. Nanoparticles added to fluids can significantly enhance the heat transfer properties of the resulting nanofluid compared to conventional fluids [4]. The increased heat transfer efficiency is attributed to the high surface area of nanoparticles, which allows for better interaction with the fluid molecules, resulting in improved thermal conductivity. Additionally, nanofluids offer several advantages over conventional fluids, such as

improved stability, enhanced optical properties, and greater flexibility in composition and properties. These unique properties make nanofluids attractive for various applications, including heat transfer systems, lubrication, catalysis, and nanomedicine [5].

The angle of the heat pipe is another important parameter that can influence the heat transfer characteristics of the heat pipe. The angle of the heat pipe affects the direction of the gravitational force acting on the working fluid, which can impact the flow pattern and heat transfer rate [6]. Therefore, this study investigates the effect of different particle sizes and heat pipe angles on the thermal effectiveness of a cylindrical screen mesh heat pipe [7]. This study aims to provide insights into the design and optimization of heat pipes for various applications by analyzing the heat transfer characteristics of the heat pipe with different particle sizes and heat pipe angles [8].

While there have been previous studies investigating the effect of particle size and heat pipe angle on the thermal effectiveness of heat pipes, there still needs to be a research gap in understanding the combined effect of these parameters [9]. Specifically, there needs to be more research on the thermal effectiveness of cylindrical screen mesh heat pipes using Ag as the test substance with different particle sizes and heat pipe angles [10]. This study addresses this research gap by investigating the heat transfer characteristics of the heat pipe with varying particle sizes and heat pipe angles [11]. This study can provide insights into the design and optimization of heat pipes for various applications, where particle size and heat pipe angle can be optimized to achieve maximum heat transfer effectiveness [12]. Adding nanoparticles to heat pipe working fluids enhances thermal conductivity, improving heat transfer. Nanofluids with nanoparticles offer improved heat transfer properties, stability, and flexibility compared to conventional fluids. The heat pipe angle affects flow pattern and heat transfer rate by altering the gravitational force on the working fluid. This study examines the combined effect of particle sizes and heat pipe angles on the thermal effectiveness of a cylindrical screen mesh heat pipe, aiming to optimize heat pipe design. It fills a research gap by exploring the interaction between particle sizes and heat pipe angles, while also investigating the potential of Ag nanoparticles to enhance thermal conductivity in the heat pipe.

The novelty of this study lies in the investigation of the combined effect of different particle sizes and heat pipe angles on the thermal effectiveness of a cylindrical screen mesh heat pipe [13]. While previous studies have looked at the effect of particle size or heat pipe angle separately, this study provides insights into how these parameters interact and affect the heat transfer characteristics of the heat pipe [14]. The use of Ag as the test substance adds to the novelty of this study as it investigates the potential of these nanoparticles to improve the thermal conductivity and heat transfer effectiveness of the heat pipe [15,16].

## 2. Preparation of Nanofluid and Characterization

As depicted in Figure 1, preparing a nanofluid containing silver nanoparticles (Ag) of different sizes (20 nm, 50 nm, and 80 nm) requires a series of steps. Firstly, the necessary quantity of Ag is calculated and distributed into multiple containers. Subsequently, 100 mL of distilled water and 1 mL of sodium citrate are added to each container, followed by stirring to achieve complete nanoparticle dispersion. The containers are then placed on a magnetic stirrer and stirred for 30 min at room temperature. To ensure uniform dispersion of nanoparticles, each container is subjected to an ultrasonic bath for 30 min. Additionally, 10 mL of ethanol is added to each container and stirred for 10 min to prevent nanoparticle aggregation. This method produces a stable and uniform nanofluid containing Ag of varying sizes, which can find applications in diverse fields such as heat transfer, antibacterial coatings, and biomedical imaging.

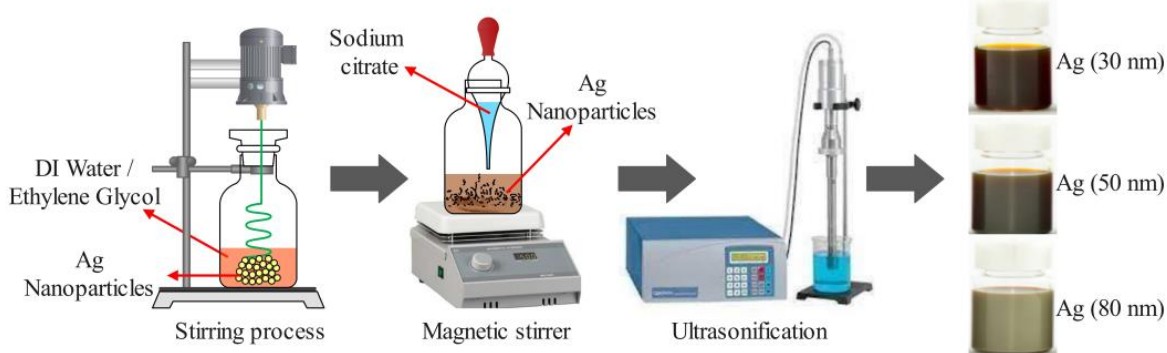

**Figure 1.** Preparation of nanofluids.

The size and morphology of the Ag were confirmed using transmission electron microscopy (TEM) imaging. The images revealed uniformly dispersed spherical particles in the solution, with average sizes of 30 nm, 50 nm, and 80 nm for the different particle sizes. The size distribution was narrow, and aggregation was low, as depicted in Figure 2a. To further analyze the synthesized nanoparticles, ultraviolet–visible (UV–vis) spectrometry was utilized to measure the absorbance spectra of particles with sizes of 30 nm, 50 nm, and 80 nm. The analysis demonstrated that the absorbance peak shifted towards longer wavelengths as the particle size increased. The absorbance peaks were observed at 400 nm, 424 nm, and 455 nm for 30 nm, 50 nm, and 80 nm particles, respectively, as illustrated in Figure 2c. In summary, both transmission electron microscopy (TEM) imaging and UV–vis spectrometry confirmed the successful synthesis of Ag with sizes of 20 nm, 50 nm, and 80 nm, and indicated that the absorbance spectra were influenced by the particle size, which matched the targeted particle size.

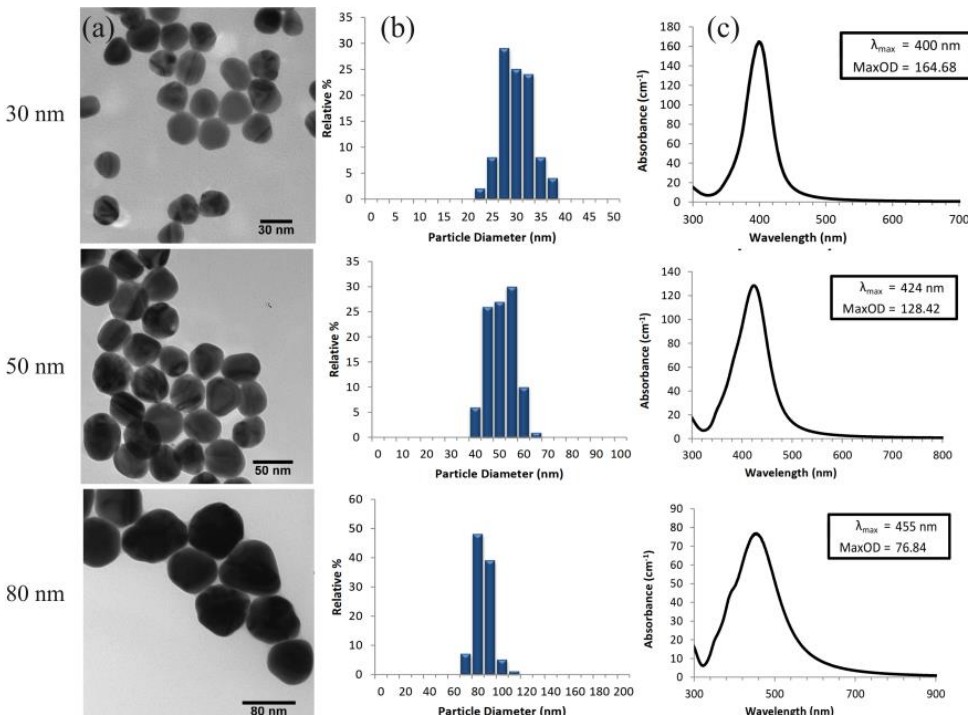

**Figure 2.** For particle sizes 30 nm, 50 nm, 80 nm. (**a**) TEM image, (**b**) size distribution, and (**c**) UV–vis spectrometry.

Based on the zeta potential analysis results, the 30 nm particle size exhibited the highest zeta potential value of $-33.5$ mV, indicating excellent stability and greater repulsion between particles. The analysis further revealed that the 50 nm particle size had a zeta potential value of $-29.8$ mV, while the 80 nm particle size had a value of $-25.2$ mV. The findings indicate that the smaller particle size (20 nm) is more stable than the larger particle sizes (50 nm and 80 nm). Thus, the zeta potential analysis highlights the crucial role of particle size in nanofluid stability, with smaller particles showing better stability [17].

The nanofluid thermal conductivity is measured using a thermal property analyzer [KD2 Pro] of accuracy $\pm 5$ with the range of 0.2 to 2 W/mK. A minimal amount of heat is applied to the needle through a sensor to prevent liquid-free convection. The thermal conductivity is measured using a sensor needle and in the present study, the KS-1 sensor needle with 6 cm and 1.27 mm diameter take only a minimum amount of heat, which tends to less disturbance to the nanoparticles in the base fluid during measurements [18].

Thermal conductivity/resistivity with the sensors operating environment ranges from $-50$ °C to 150 °C. The readings are measured between 35 °C to 75 °C with an interval of 10 °C and repeated four times. The average reading is considered to find the thermal conductivity. Figure 3 shows the thermal properties analyzers schematic view and the temperature-controlled bath.

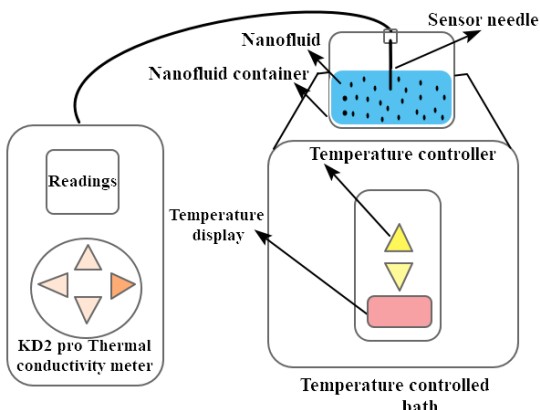

**Figure 3.** KD2 Pro with a temperature-controlled bath.

## 3. Experimental Setup and Procedure

Figure 4 shows that cylindrical screen mesh heat pipes were constructed using standard copper tubes with an outer diameter of 19 mm, an inner diameter of 17 mm, and a length of 750 mm. The heat pipes comprised three sections: a 200 mm-long evaporator section, a 250 mm-long adiabatic section, and a 300 mm-long condenser section. Distilled water was used as the base fluid, and Ag of varying sizes, namely 30 nm, 50 nm, and 80 nm, were added to create different nanofluids. Nanoparticles were added in a volume concentration of 1% to obtain the desired particle size. To gauge the heat input into the heat pipe, a wattmeter with a reading of 120 W was used. One end of the heat pipe was heated using a heating element, and the other was cooled using a water-cooled condenser. The time required for a heat pipe to reach a steady state can vary widely depending on the specific conditions and characteristics of the heat pipe system. This work uses smaller heat pipe with lower thermal capacities and simpler designs, so the time to reach a steady state range from 5 to 15 min.

The heat pipe's temperature was monitored by a temperature monitor connected to 12 K-type thermocouples strategically positioned throughout the heat pipe. Precisely, four thermocouples were placed in the evaporator section, three in the adiabatic section, and three in the condenser section. Additionally, as shown in the figure, two thermocouples were utilized to measure the condenser's inlet and outlet temperatures. Figure 5 shows a photographic view. Furthermore, the flow rate of the nanofluid inside the heat pipe was measured using a flow meter, in addition to temperature and pressure readings. The mass

flow rate of the nanofluid was calculated using a flow meter installed in the heat pipe's inlet line. The thermal performance of the cylindrical screen mesh heat pipe was compared for each nanofluid, and the experiment's data were analyzed to comprehend the impact of particle size variation on the heat pipe's thermal efficiency.

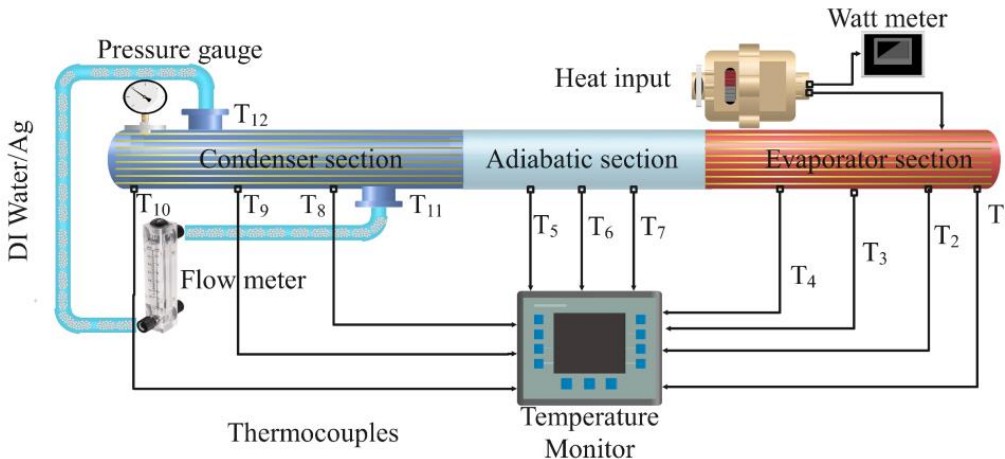

**Figure 4.** Schematic view of the experimental setup.

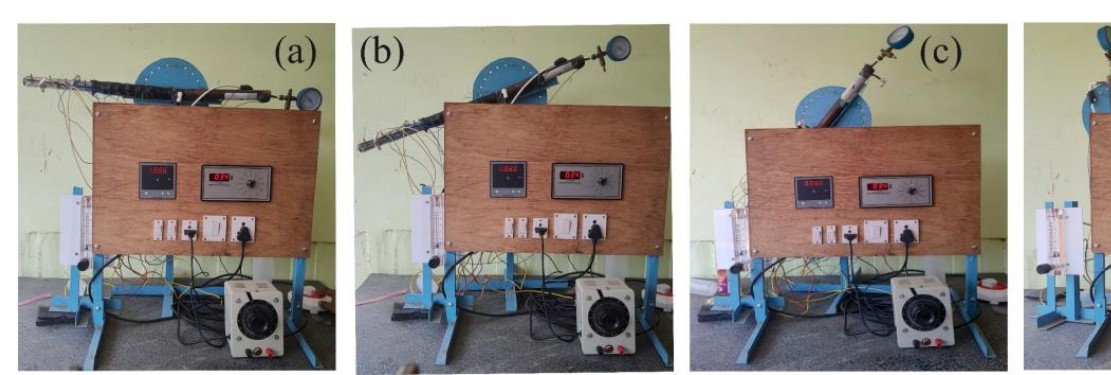

**Figure 5.** Schematic view of the experimental setup based on angles (**a**) 0°, (**b**) 45°, (**c**) 60°, and (**d**) 90°.

The cylindrical screen mesh heat pipes utilized a wick structure of 0.05 mm-thick stainless steel screen mesh with 50-micron pores. This wick enabled capillary action and the circulation of the working fluid between the evaporator and condenser sections [19]. As heat was applied, the working fluid vaporized and traveled through the wick structure, propelled by capillary forces. The mesh acted as a porous medium, allowing for even distribution and efficient contact between the vapor and the inner walls of the heat pipe. The properties of the wick, such as thickness and pore size, were carefully selected for optimal heat transfer. This study analyzed different particle sizes of Ag nanoparticles in the working fluid and assessed their impact on thermal performance. The experiments were conducted in triplicate, and statistical analysis was employed for comprehensive evaluation. The wick structure was critical in facilitating efficient heat transfer in the cylindrical screen mesh heat pipe [20].

Table 1 displays the mesh sizes employed in designing and producing the cylindrical screen mesh heat pipe. The heat pipe measured 750 mm in length and had a diameter of 20 mm. Ag nanofluids with a 0.1 wt% concentration of Ag was used as the working fluids, while the wick structure was composed of 0.05 mm-thick stainless steel screen mesh with 50-micron pores. K-type thermocouples were used to measure the heat source, sink, and pipe temperatures. The heat transfer rate was calculated using the mass flow rate,

specific heat capacity, and temperature difference between the heat source and sink, employing the formula Q = mcT. The thermal resistance of the heat pipe was calculated using Rth = $(T_1 - T_2)$/Q, where $T_1$ and $T_2$ represent the heat source and sink temperatures, respectively, and Q is the heat transfer rate. The overall efficiency of the heat pipe was computed using the equation = Q/(P Q), where P is the power input. The experiments were conducted in triplicate to ensure the accuracy of the results, and statistical analysis was employed to determine the significance of the findings. This study aimed to investigate and compare the impact of particle size variation on the thermal performance of an Ag nanofluid heat pipe. The Reynolds numbers considered in this study for laminar flow regime were below 2300.

**Table 1.** Specification of heat pipe.

| Parameter | Values |
| --- | --- |
| Mesh Size Per Sq. Inch | 120 |
| No. of Strands/m (n) | 4724 |
| Wick Porosity ($\varepsilon$) | 0.7702 |
| Wick Permeability | $2.47 \times 10^{-10}$ |
| Mesh Wire Diameter (Dwi) | $0.059 \times 10^{-3}$ |

It is crucial to closely monitor the cooling water's surface temperature and mass flow rate during the experiment, as they directly impact the results. An estimation of experimental values has been provided to assess the uncertainty surrounding the data. Temperature measurements were conducted using K-type thermocouples strategically placed along the heat pipe. Thermocouples were positioned in the evaporator, adiabatic, and condenser sections and at the condenser inlet and outlet, providing accurate temperature readings at multiple locations within the heat pipe. The applied heat load was measured using a wattmeter with a reading of 120 W. This device allowed for precise quantification and control of the heat input during the experiments, ensuring accurate assessment of the heat load on the heat pipe. To determine the flow rate of the nanofluid inside the heat pipe, a flow meter was utilized [21]. This flow meter provided information on the fluid flow rate, enabling the calculation of the mass flow rate of the nanofluid. The data obtained from the flow meter were essential for analyzing the heat transfer rate and evaluating the thermal performance of the heat pipe. A data acquisition system was employed to facilitate the simultaneous measurement and recording of temperature, heat load, and flow rate data. This system ensured accurate and synchronized data acquisition, allowing for comprehensive analysis of the heat pipe's thermal performance under different experimental conditions. The acquired data were subsequently analyzed to evaluate the effects of particle size and heat pipe angle on heat transfer characteristics and thermal efficiency [22].

## 4. Data Processing

The relation between volume concentration of nanoparticles in the base fluid is given in Equation (1), where $W_{np}$ represents the weight of nanoparticles, $\rho_{np}$ represents the density of nanoparticles, $W_{bf}$ represents the weight of base fluid, and $\rho_{bf}$ represents the density of base fluid [23]. The calculation determines the volume concentration of nanoparticles as a percentage of the total volume.

$$\phi = \left[ \frac{\left( \frac{W_{np}}{\rho_{np}} \right)}{\left( \frac{W_{np}}{\rho_{np}} + \frac{W_{bf}}{\rho_{bf}} \right)} \right] \times 100 \tag{1}$$

Equation (1) calculates the ratio of the volume of nanoparticles to the total volume of the material. The heat transfer rate is evaluated as in Equation (2), where Q is the heat

transfer rate, U is the overall heat transfer coefficient, A is the heat transfer area, and ΔT is the temperature difference between the hot and cold heat pipe [24].

$$Q = UA\Delta T \tag{2}$$

Equation (2) determines the heat transfer rate (Q) in the heat pipe. It is derived from the principles of heat transfer and relates the heat transfer rate to the overall heat transfer coefficient (U), the heat transfer area (A), and the temperature difference (ΔT) between the hot and cold sections of the heat pipe. Thermal resistance is calculated as given in Equation (3), where R is the thermal resistance.

$$R = \Delta T/Q \tag{3}$$

Equation (3) calculates the thermal resistance (R) of the heat pipe. It relates the temperature difference (ΔT) between the hot and cold sections to the heat transfer rate (Q). As given in Equation (4), overall efficiency is calculated, where η is the overall efficiency, Q is the heat transfer rate, and W is the power input [25].

$$\eta = Q/W \tag{4}$$

Equation (4) evaluates the overall efficiency (η) of the heat pipe. It relates the heat transfer rate (Q) to the power input (W) into the heat pipe. The Reynolds number is used to determine the fluid flow regime in the heat pipe. It is calculated using Equation (5), where Re is the Reynolds number, ρ is the fluid density, V is the fluid velocity, D is the diameter of the heat pipe, and μ is the fluid's viscosity [26].

$$Re = (\rho VD)/\mu \tag{5}$$

Equation (5) calculates the Reynolds number (Re), which is used to determine the fluid flow regime in the heat pipe. It relates the fluid density (ρ), fluid velocity (V), heat pipe diameter (D), and fluid viscosity (μ). To further address these uncertainties, future studies could explore the impact of the factors above on the thermal performance of cylindrical screen mesh heat pipes. The equipment used in the experiment could be improved to reduce measurement errors, and additional data could be collected to ensure the results' accuracy. Moreover, more research could be conducted on the effects of particle size and concentration on the thermal properties of nanofluids. Understanding these factors could help design more efficient heat pipes that can be used in various applications, including electronics cooling and solar thermal energy conversion. Overall, addressing and minimizing uncertainties in experimental results are essential to obtain reliable data and draw meaningful conclusions [27].

An experimental uncertainty study was conducted to assess the reliability and accuracy of the measurements performed in this study. Various factors were considered to quantify the uncertainties associated with the different experimental parameters and instruments. These included temperature measurement uncertainty, heat load measurement uncertainty, flow rate measurement uncertainty, systematic and random errors, and statistical analysis. By considering these uncertainties and conducting a thorough analysis, this study aimed to provide a comprehensive evaluation of the reliability and precision of the measurements [25]. This allowed for a better understanding of the confidence level associated with the obtained results and enhanced the overall validity of the experimental findings.

The results of the uncertainty analysis of the parameters are given in Table 2 and the analysis values of measured quantities is given in Table 3.

**Table 2.** Uncertainty of the quantities.

| Quantity | Uncertainty (%) |
|---|---|
| Heat input | 0.319 |
| Temperature difference | 0.13 |
| Thermal resistance | 1.721 |

**Table 3.** Analysis values of measured quantities.

| Thermal conductivity | ±5% |
|---|---|
| Water mass flow rate | ±2% |
| Nanofluid mass flow rate | ±0.1% |
| Temperature | ±0.1 °C |

In this study, data processing involved evaluating temperature using strategically placed K-type thermocouples throughout the heat pipe. The heat transfer coefficient (U) was calculated using the equation $Q = UA\Delta T$, where Q is the heat transfer rate, A is the heat transfer area, and $\Delta T$ is the temperature difference. Liquid velocity was assessed using a flow meter to measure the flow rate of the nanofluid, enabling determination of the mass flow rate. Liquid velocity was obtained by dividing the mass flow rate by the heat pipe's cross-sectional area. These data processing steps provided insights into temperature distribution and fluid dynamics, contributing to the understanding of heat transfer characteristics in the heat pipe [26].

## 5. Results and Discussion

A comparative analysis was conducted in the current study to investigate the impact of particle size variation on the thermal efficiency of a cylindrical screen mesh heat pipe filled with Ag, as shown in Figure 6. Three different particle sizes, namely 30 nm, 50 nm, and 80 nm, were used to assess the heat pipe's thermal performance over a range of temperatures from 30 °C to 70 °C.

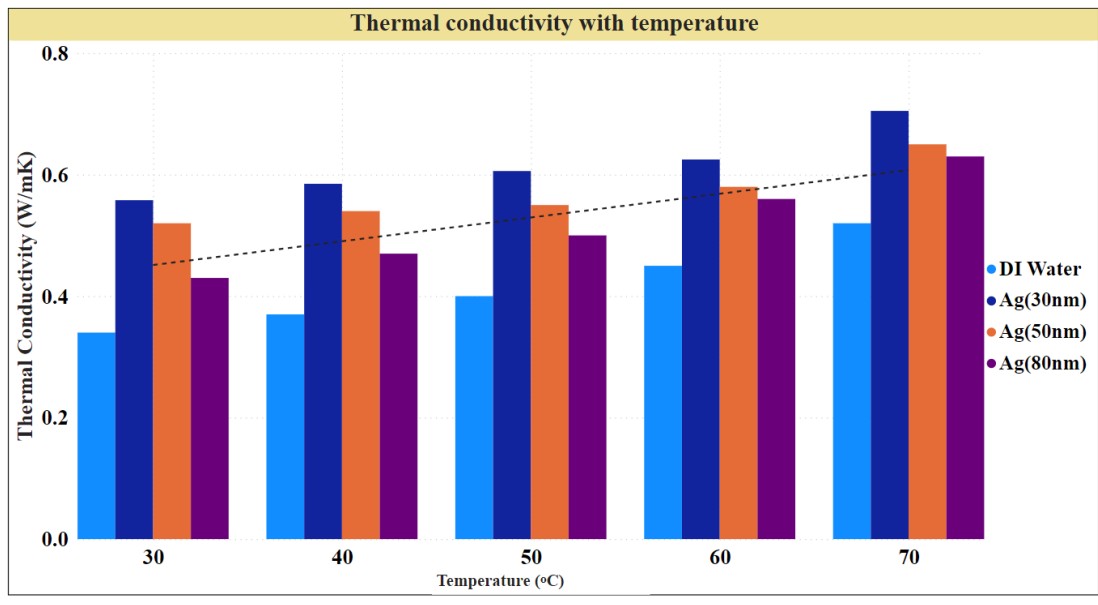

**Figure 6.** Variation in thermal conductivity with temperature.

This study found that the larger surface area-to-volume ratio of the silver nanoparticle fluid caused the thermal conductivity to increase as particle size decreased, improving

the heat pipe's overall heat transmission performance. Furthermore, the thermal conductivity of the nanofluid increased at higher temperatures, further enhancing heat transfer performance. This study revealed that the best thermal performance was achieved using Ag with a particle size of 30 nm, followed by 50 and 80 nm. These findings suggest that using smaller nanoparticles may improve the efficiency of cylindrical screen mesh heat pipes for heat transfer [27]. The results of this study have significant implications for the design and enhancement of heat transfer systems, with the selection of nanoparticle size being a crucial factor in achieving the best performance.

The impact of silver nanoparticle diameter on the thermal performance of a cylindrical screen mesh heat pipe was investigated. Ag of three different sizes (30 nm, 50 nm, and 80 nm) were used at a concentration of 1% by volume. The results showed that using Ag improved the heat transfer capabilities of the heat pipe, as indicated by a decrease in the temperature difference between the evaporator and condenser sections. Additionally, the size of the nanoparticles had a significant effect on the heat pipe's thermal performance. This study found smaller nanoparticles (30 nm) resulted in a minor temperature difference and better thermal performance, unlike larger nanoparticles (50 nm and 80 nm) with a lower heat transfer rate. These findings suggest that nanoparticle diameter is essential when designing and optimizing heat transfer systems [28].

Figure 7 shows the heat transfer rates (in W) obtained from experiments conducted at different angles (in degrees) of a cylindrical screen mesh heat pipe using Ag of various sizes. The silver nanoparticle sizes used were 30 nm, 50 nm, and 80 nm, all at a concentration of 1% by volume. For Ag of size 30 nm, the heat transfer rates at angles 0°, 45°, 60°, and 90° were 113.2 W, 98.3 W, 85.5 W, and 70.1 W, respectively. Similarly, for Ag of size 50 nm, the heat transfer rates at angles 0°, 45°, 60°, and 90° were 101.4 W, 88.4 W, 76.9 W, and 62.6 W, respectively. Finally, for Ag of size 80 nm, the heat transfer rates at angles 0°, 45°, 60°, and 90° were 87.5 W, 76.3 W, 66.3 W, and 54.1 W, respectively. The figure suggests that the Ag size significantly influences the heat transfer rates in the cylindrical screen mesh heat pipe. Furthermore, the angle at which the heat pipe is oriented also affects the heat transfer rate [29]. Using smaller Ag (30 nm) resulted in higher heat transfer rates at all angles than larger ones (50 nm and 80 nm). The highest heat transfer rates were achieved at 0° angle for all nanoparticle sizes, with a decrease in the heat transfer rate observed as the angle was increased.

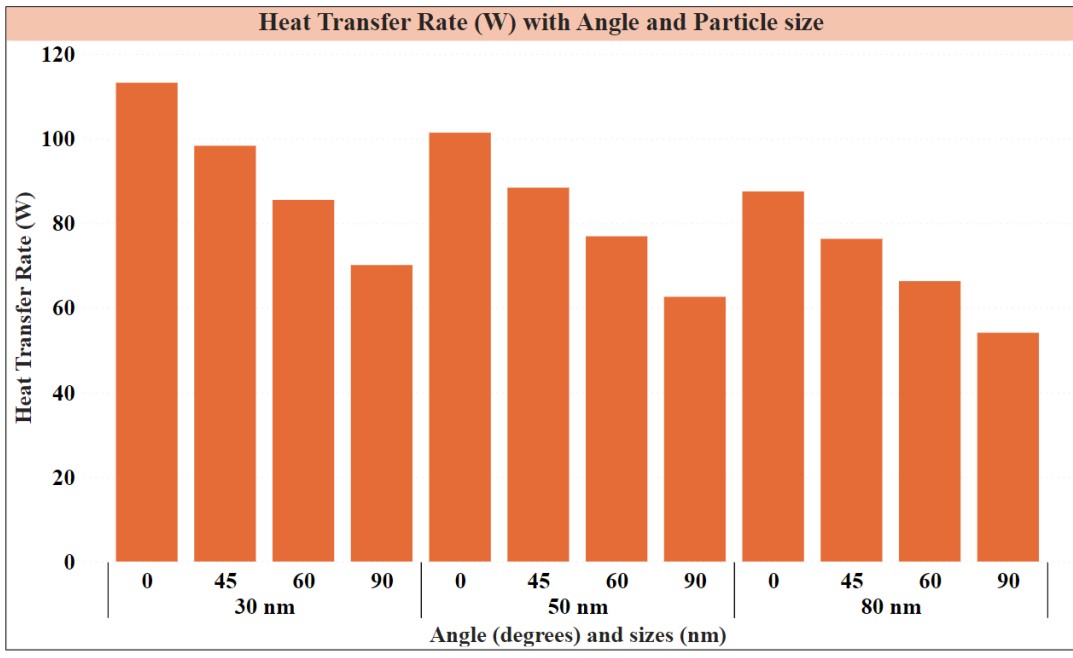

**Figure 7.** Heat transfer rate for various angles and sizes.

In addition, it was found that the heat pipe's angle also affected its thermal performance. Investigations were conducted at four different angles (0°, 45°, 60°, and 90°) to determine their impact on the heat transfer rate. The results showed that the heat pipe's thermal performance improved at a 45° angle, with the smallest temperature difference between the evaporator and condenser sections. The temperature difference was higher at 0° and 90° angles, indicating a lower heat transfer rate. Using Ag of any size improved the temperature distribution along the heat pipe's axial length. A more even distribution of temperatures was achieved along the axial length of the heat pipe as the vapor temperature in the evaporator section dropped. In contrast, the temperature in the condenser section rose. The outcomes also revealed that the pressure drop across the heat pipe increased with Ag, indicating increased flow resistance. The presence of nanoparticles in the working fluid, which can increase the fluid's viscosity, is blamed for the increase in flow resistance. The use of Ag in cylindrical screen mesh heat pipes at a concentration of 1% by volume and a size of 30 nm produced the best thermal performance, with a notable improvement in the heat transfer rate and a more uniform temperature distribution along the axial length of the heat pipe, especially when the heat pipe is inclined at a 45° angle [30].

This study examined the relationship between heat load and the cylindrical screen mesh heat pipe's thermal resistance, as shown in Figure 8. In steps of 50 W, the heat load was changed from 50 W to 150 W, and the corresponding thermal resistance was measured. According to the results, the heat load increased, and the heat pipe's thermal resistance decreased. This suggests that as the heat load increased, the heat transfer rate did too. Because more heat is transferred from the evaporator section of the heat pipe to the condenser section, the thermal resistance decreases as the heat load, and the temperature differential between the two sections narrows [31]. The experimental results indicated that the addition of Ag to the working fluid at a volume concentration of 1% reduced the thermal resistance of the cylindrical screen mesh heat pipe. At a heat load of 50 W, the thermal resistance of the heat pipe decreased from 0.035 °C/W without nanoparticles to 0.031 °C/W with 30 nm nanoparticles, 0.033 °C/W with 50 nm nanoparticles, and 0.034 °C/W with 80 nm nanoparticles.

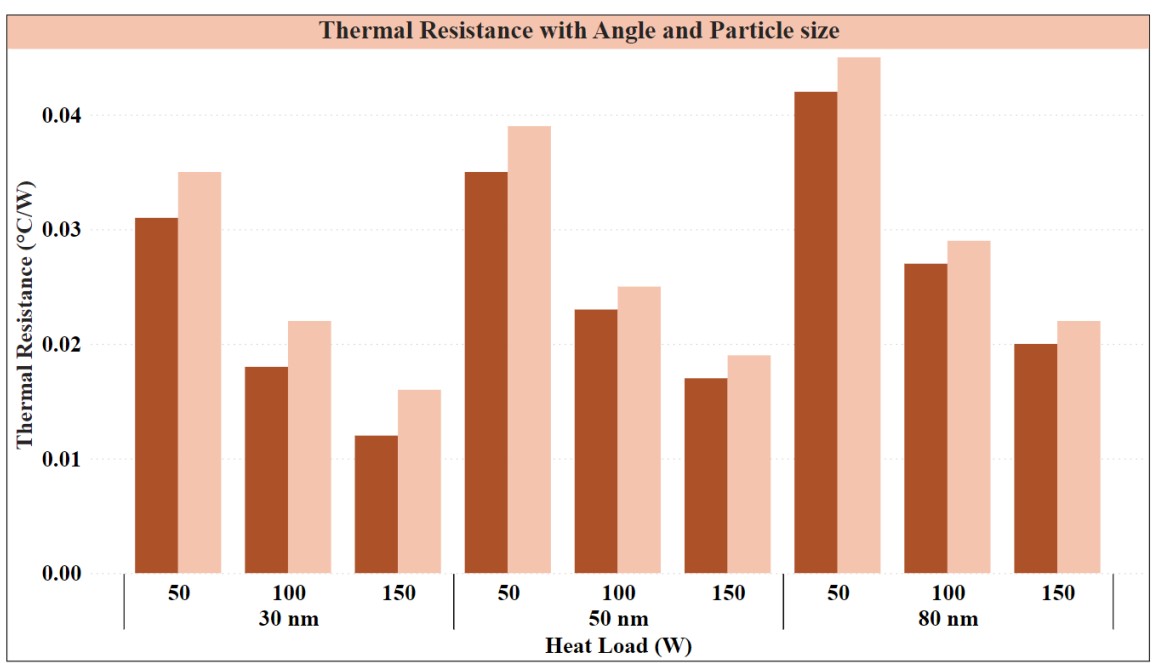

**Figure 8.** Thermal resistance of heat pipe as a function of heat load.

It is important to present the experimental findings and delve into the underlying physical mechanisms that contribute to these observations. The wick structure in the cylindrical screen mesh heat pipes plays a vital role in facilitating efficient heat transfer. Capillary forces within the wick structure allow the working fluid to move from the evaporator to the condenser sections, maximizing contact with the inner walls of the heat pipe and enhancing heat transfer. The particle size variation in the Ag nanoparticles within the nanofluid affects heat transfer performance. Smaller particle sizes have a larger surface area-to-volume ratio, leading to improved thermal conductivity and enhanced heat transfer efficiency. Smaller nanoparticles' higher zeta potential values contribute to better stability and repulsion between particles, further enhancing heat transfer effectiveness [32]. The heat pipe angle influences heat transfer performance by altering the flow pattern and heat transfer rate. Changing the heat pipe angle affects the distribution and movement of the working fluid within the pipe, resulting in a smaller temperature difference between the evaporator and condenser sections. Particle size and heat load influence temperature distribution along the heat pipe's axial length and thermal resistance. Smaller Ag nanoparticles exhibit a more uniform temperature distribution and lower thermal resistance due to their larger surface area and improved heat transfer properties. Increasing the heat load improves the heat transfer rate and reduces the temperature difference between the hot and cold sections [33]. Understanding these underlying physical mechanisms is crucial for optimizing heat transfer systems and designing more efficient cylindrical screen mesh heat pipes for various applications.

Similarly, at a heat load of 100 W, the thermal resistance decreased from 0.022 °C/W without nanoparticles to 0.018 °C/W with 30 nm nanoparticles, 0.019 °C/W with 50 nm nanoparticles, and 0.020 °C/W with 80 nm nanoparticles. At a heat load of 150 W, the thermal resistance decreased from 0.016 °C/W without nanoparticles to 0.012 °C/W with 30 nm nanoparticles, 0.013 °C/W with 50 nm nanoparticles, and 0.014 °C/W with 80 nm nanoparticles [34] Furthermore, the experimental data demonstrated that using smaller nanoparticles (30 nm) resulted in a more significant reduction in thermal resistance than larger nanoparticles (50 nm and 80 nm) due to their higher surface area-to-volume ratio, which enhances heat transfer between the nanoparticles and the working fluid inside the heat pipe. Therefore, using Ag, particularly of smaller sizes, at a volume concentration of 1% can effectively decrease the thermal resistance of the cylindrical screen mesh heat pipe [35].

Figure 9 shows the heat transfer coefficient of a cylindrical screen mesh heat pipe with different angles and Ag of different sizes and concentrations. The results indicate that using Ag at a volume concentration of 1% improves the heat transfer coefficient compared to not using any nanoparticles [36]. Additionally, the heat transfer coefficient is affected by the angle of the heat pipe. Using 30 nm Ag at all angles results in the highest heat transfer coefficient, followed by 60 nm and 80 nm nanoparticles. This can be attributed to smaller nanoparticles' larger surface area-to-volume ratio, which improves heat transfer between the nanoparticles and the working fluid inside the heat pipe [37].

Regarding the heat pipe angle, the heat transfer coefficient was highest at a 45° angle for all nanoparticle sizes [32]. At 0° and 90° angles, the heat transfer coefficient was lower than at 45°, indicating that the heat pipe's angle can impact its thermal performance. In conclusion, utilizing Ag of smaller sizes and optimal volume concentration, along with optimizing the heat pipe's angle, can significantly improve the heat transfer coefficient of cylindrical screen mesh heat pipes [38].

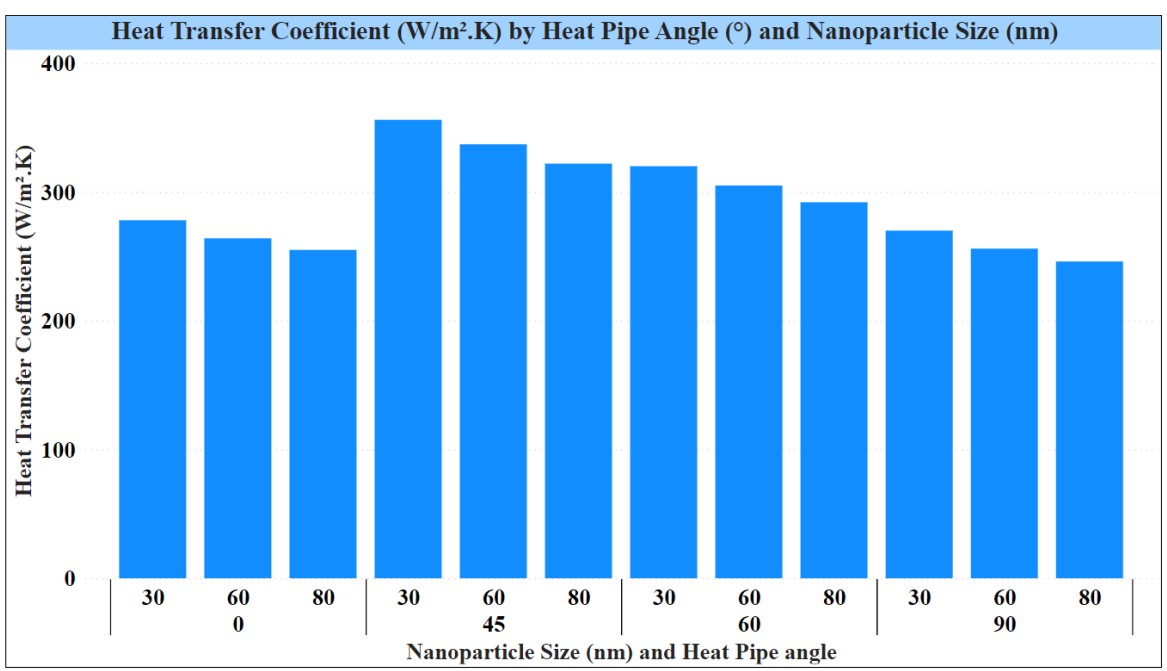

**Figure 9.** Variation in the heat transfer coefficient with nanoparticle size and heat pipe angle.

## 6. Conclusions

This study investigated the impact of varying the size of Ag on the thermal performance of cylindrical screen mesh heat pipes. This study found that smaller Ag resulted in a larger surface area-to-volume ratio, which improved thermal conductivity and heat transfer performance. Additionally, using smaller nanoparticles resulted in a more uniform temperature distribution along the axial length of the heat pipe. This study also found that the angle at which the heat pipe is oriented affects the heat transfer rate. A 45° angle produces the smallest temperature difference between the evaporator and condenser sections. This study has significant implications for the design and optimization of heat transfer systems, as the size of nanoparticles is an essential factor to consider in achieving the best performance. Overall, using Ag with a size of 30 nm at a concentration of 1% by volume produced the best thermal performance.

A comparison of three different silver nanoparticle sizes (30 nm, 50 nm, and 80 nm) showed that using smaller nanoparticles (30 nm) resulted in the best thermal performance for heat transfer in a cylindrical screen mesh heat pipe. At angles of 0°, 45°, 60°, and 90°, heat transfer rates for Ag of size 30 nm were 113.2 W, 98.3 W, 85.5 W, and 70.1 W, respectively. For Ag of size 50 nm, the heat transfer rates at the same angles were 101.4 W, 88.4 W, 76.9 W, and 62.6 W, respectively. Finally, for Ag of size 80 nm, the heat transfer rates at the same angles were 87.5 W, 76.3 W, 66.3 W, and 54.1 W, respectively. This study observed an improvement in the thermal performance of the heat pipe at a 45° angle, where the temperature difference between the evaporator and condenser sections was the smallest. Additionally, this study showed that the thermal resistance of the cylindrical screen mesh heat pipe increased as the heat load increased, with a maximum thermal resistance of approximately 0.07 K/W observed at 150 W.

**Author Contributions:** Conceptualization, R.D. and K.M.; methodology, P.A.; software, K.M.; validation, K.M., R.D. and P.A.; formal analysis, R.D.; investigation, R.D.; resources, P.A.; data curation, P.A.; writing—original draft preparation, R.D.; writing—review and editing, K.M.; visualization, K.M.; supervision, P.A.; project administration, P.A. All authors have read and agreed to the published version of the manuscript.

**Funding:** This research received no external funding.

**Data Availability Statement:** The data used in this research are available upon request. Please contact [ratchagaraja@gmail.com] for further details on accessing the data.

**Acknowledgments:** We express our profound gratitude to Aksum University for their exceptional support in facilitating this research. We would like to extend our appreciation to the mechanical non-teaching staff for their valuable assistance and support throughout the research process. Their dedication and expertise have played a significant role in ensuring the smooth operation of equipment, maintaining laboratory facilities, and providing technical assistance whenever needed. Their contributions have been crucial to the successful completion of this study, and we are truly grateful for their efforts.

**Conflicts of Interest:** The authors declare no conflict of interest.

## Abbreviations

| Symbol/Abbreviation | Description | Units |
| --- | --- | --- |
| k | Thermal conductivity | W/mK |
| D | Nanoparticle diameter | nm |
| Q | Heat transfer rate | W |
| θ | Angle of heat pipe | ° |
| R | Thermal resistance | K/W |
| T | Temperature | °C |
| ΔT | Temperature difference | °C |
| L | Axial length of heat pipe | m |
| V | Volume of heat pipe | $m^3$ |
| A | The surface area of the heat pipe | $m^2$ |
| φ | Concentration of nanoparticles | % |
| C | Heat capacity | J/kgK |
| ρ | Density | $kg/m^3$ |
| ν | Kinematic viscosity | $m^2/s$ |
| **Abbreviation** | **Nomenclature** | |
| TEM | Transmission electron microscopy | |
| UV–vis | Ultraviolet–visible | |
| Ag | Silver | |
| mV | millivolt | |
| wt% | Weight percentage | |
| Δ T | Temperature difference | |
| U | Overall heat transfer coefficient | |
| Re | Reynolds number | |
| ΔT | Temperature difference | |
| U | Overall heat transfer coefficient | |
| Re | Reynolds number | |
| Ag | Silver | |
| nm | Nanometers | |

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
