# Peer review of "Nanoparticle Size and Heat Pipe Angle Impact on the Thermal Effectiveness of a Cylindrical Screen Mesh Heat Pipe"

_2673-3161, doi:10.3390/applmech4030045_

Round 1

Reviewer 1 Report

some revisions must be done

1- please valid your work

2-Please provide the TEM of nanofluid

3-Please add the calibration, sensitivity analysis

4-The intro is poor please add some new related papers in your work

5-Please highlight the novelty

6-more physical explanation must be added to paper  

Please overall check the paper against typos  error

Reviewer 2 Report

The authors, in this paper, investigated the effects of particle size and heat pipe angle on the thermal effectiveness of a cylindrical screen mesh heat pipe using silver nanoparticles (Ag) as the test substance. Some important assumptions seem to be missing in this paper. However, it would be interesting if the authors could improve the paper once again using the following comments:

 1.      The study used the experimental method to analyze the effects of particle size and heat pipe angle on the thermal effectiveness of a cylindrical screen mesh heat pipe. In this study, what novelty is using of nanoparticles Ag have? The analysis is based on three different particle sizes (30nm, 50nm, and 80nm) and four different heat pipe angles (0°, 45°, 60°, and 90°). These values can be employed in general. Also, previous literatures already show the characteristics of heat transmission characteristics of the heat pipe. If so, can't we just use the results of previous research on them without newly analyzing it?

2.      Is it necessary to take concentration of 1% by volume. Whether it can be increased or reduced for the problem under consideration.

3.      What are the problem parameters and why you have chosen such range of parameters. 

4.      The authors focused mainly on the impact of silver nanoparticles diameter on the thermal performance. Whereas the other prominent features are missing. Please correlate your results with the other experimental ones even in limiting cases. It will validate your experimental results. Also cite some recent papers for the thermal efficiency using different nanoparticles and different methodologies. For instance. (i) Impact of magnetic field localization on the vortex generation in hybrid nanofluid flow. (ii) A case study of different magnetic strength fields and thermal energy effects in vortex generation of Ag-TiO2 hybrid nanofluid flow (iii) Thermal case study and generated vortices by dipole magnetic field in hybridized nanofluid flowing: Alternating direction implicit solution (iv) Numerical Assessment of Dipole Interaction with the Single-Phase Nanofluid Flow in an Enclosure: A Pseudo-Transient Approach (v) Numerical study of Lorentz force interaction with microstructure in channel flow. (vi) Prediction of new vortices in single-phase nanofluid due to dipole interaction (vii) Molecular Interaction and Magnetic Dipole Effects on Fully Developed Nanofluid Flowing via a Vertical Duct Applying Finite Volume Methodology.

5.      What is the role of heat load (increase/decrease) on the thermal resistance of the cylindrical screen mesh heat pipe. Discuss in the revise paper.

6.      Physically interpret the section 4 of the paper “Data processing”. e. g. provide detail for the governing equations (1)-(5). 

Reviewer 3 Report

The authors presented an experimental study on the Nanoparticle Size and Heat Pipe Angle Impact on Thermal Effectiveness of a Cylindrical Screen Mesh Heat Pipe.

The main quantitative findings are to be mentioned in the abstract.

The introduction is relatively short and may be extended.

In figure 2, the nanoparticles sizes presented at the first row, should be at the first left column.

Have you checked the stability of the nanofluid?

More details on the measurement techniques and data acquisitions system are to be provided.

What is the needed time to reach to steady state?

A detailed experimental uncertainty study is to be performed.

The unit of temperature is to be added in Fig 5.

How is the thermal conductivity measured? The used equipment is to be presented and a photo is to be added.

What is the considered range of Re?

The following paper may be added to the literature review:

10.3390/app122211614

English level needs to be improved.

English level needs to be improved.

Reviewer 4 Report

The authors have investigated the angle inclination of the pipes and its effect on heat transfer. Pipe size is an important parameter in this study. In microtubes, the influence of gravity is negligible. I recommend discussing the channel classification in the introduction

The wick structure should be described in more detail. I recommend showing it in Figure 3 and providing a thorough description in the text..

All abbreviations should be described in the text and included in the nomenclature. Please pay particular attention to paragraph 3.

The data processing should be described in more detail. Specifically, it would be helpful to know which thermocouples were used for data evaluation, how the heat transfer coefficient was calculated, and how the liquid velocity was evaluated.

volume concentration of nanoparticles 
Ï• = (m
p/Vf) '100%

This is the dimensional value. Please check it.

In the discussion, it is necessary to describe not only the facts found in the experiment, but also the physical mechanisms that cause them.

English needs to be improved.

Round 2

Reviewer 1 Report

Dear Prof

The paper is improved and can be published in current form

Regards

-

Author Response

Thank you for your valuable feedback and encouraging review. Your positive assessment motivates us to enhance our work further. We sincerely appreciate your time and consideration and are grateful for the opportunity to contribute our research to the scientific community.

Reviewer 2 Report

The paper has been revised significantly according to the comments. I recommend the revised version of the paper for publication.

Author Response

(The authors gave the same response as above.)

Reviewer 3 Report

Accept in present form

Author Response

(The authors gave the same response as above.)

Reviewer 4 Report

The ratio mp/Vf has the dimension kg/m3.

Ï• must bea dimensionless value. Revise it.

-
